# Navigating antiretroviral adherence in boarding secondary schools in Nairobi, Kenya: A qualitative study of adolescents living with HIV, their caregivers and school nurses

**Nicholas Kipkurui** [1]*, **Emmah Owidi** [2], **James Ayieko** [3], **Gerald Owuor** [1],
**Irene Mugenya** [4], **Kawango Agot** [1], **Alison C. Roxby** [5]

1 Impact Research Development Organization, Kisumu, Kenya, 2 Kenya Medical Research Institute–Center for Clinical Research, Partners in Health Research and Development, Nairobi, Kenya, 3 Kenya Medical Research Institute–Center for Microbiology Research, Kisumu, Kenya, 4 Kenya Medical Research Institute–Wellcome Trust Research Programme, Kilifi, Kenya, 5 University of Washington, Departments of Medicine, Global Health, and Epidemiology, Seattle, WA, United States of America

* kipkuruinicholas@gmail.com

**Data Availability Statement:** The semi-structured interview guide and the codebook used to collect

## Abstract

In Kenya, adolescents spend much of their formative years in boarding secondary schools, which presents a challenging environment for antiretroviral (ART) adherence support among adolescents living with HIV (ALHIV). We examined the experiences of ALHIV, caregivers of adolescents, and school nurses regarding navigating ART adherence in boarding secondary schools. Between July and November 2022, we conducted focus group discussions (FGDs) among ALHIV attending boarding schools in Nairobi, Kenya, and caregivers of ALHIV, and in-depth interviews (IDIs) with school nurses. Clinic records were used to identify ALHIV and caregivers, who were invited to participate based on their availability. We categorized boarding schools into national, county, and sub-county levels and selected two schools from each category. We obtained permission from head teachers and invited school nurses to take part in virtual IDIs. The interviews were audio-recorded, transcribed verbatim, and analyzed thematically. We conducted two FGDs with 11 caregivers, two FGDs with 18 adolescents, and 7 IDIs with school nurses. Most of the ALHIV reported having disclosed their HIV status to a school nurse or teacher during admission. School nurse friendliness, being understanding, fair, and confidential were qualities associated with ALHIV willingness to confide in them. Strategies ALHIV used to adhere to medication included: waiting until students were engaged in other activities, waking up early, stepping away from others, and stating their drugs were for different ailments. Caregivers were nervous about school-based adherence counseling, fearing it could lead to inadvertent disclosure of adolescents' HIV status and stigmatization by fellow students. All school nurses reported lacking appropriate training in HIV adherence counseling for adolescents. ALHIV have devised innovative strategies to navigate pill-taking and enlist quiet support while operating in stigmatized school environments. Establishment of a strong school nurse-adolescent rapport and building nurses' skills are key to improving school-based support for ALHIV.

and analyze the qualitative interviews are included as additional files. The full transcripts of the qualitative interviews may be available after consultation with the KEMRI Scientific and Ethics Review Unit, as per the ethical approval obtained for this research. For data sharing the following person can be contacted. Name: Ivy Wango Obare P.O BOX 9171-40141 Impact Research and Development Organization contact +254751237613 email:ivywangobare@gmail.com organization URL:https://www.impact-rdo.org.

**Funding:** This work was supported by Bill & Melinda Gates Foundation [OPP1213352/ INV009408 to KA] through the Reducing HIV in Adolescents and Youths (RHAY) mentorship initiative and the University of Washington Center for AIDS Research (P30 AI027757 to AR). For open access, the author has applied a CC-BY public copyright license to any author-accepted manuscript version arising from this submission. The funders had no role in study design, data collection and analysis, decision to publish, or preparation of the manuscript.

**Competing interests:** The authors have declared that no competing interests exist.

## Introduction

Adolescents and young people aged 10 to 24 years constitute a third of people living with HIV worldwide and experience the worst antiretroviral outcomes of any age group [1]. Progress in improving HIV outcomes for children and adolescents is lagging behind other age groups, particularly in low resource countries [2,3]. School-going children living with HIV are negatively impacted by school environments due to bullying, inadvertent serostatus disclosure, stigma, and antiretroviral therapy (ART) adherence challenges [4].

A recent systematic review of existing research to improve ART adherence for youth in low- and middle-income countries revealed scarce proven strategies in this age group [5,6]. Some strategies, including youth-tailored care [7], peer support networks [8,9], and education-based interventions [10], are all linked to improved antiretroviral adherence and retention rates. According to Kenya's Fast-track Plan to end HIV and AIDS among adolescents and young people [11], school-based follow-up by a community health worker, adolescent support groups, and school-based health literacy programs are essential interventions to promote retention in HIV care for adolescents in school. Other researchers have similarly concluded that adolescents living with HIV need a supportive environment and access to the right services in order to enhance their psychological well-being and protect them from the negative consequences of living with HIV [12]. Research in Rwanda demonstrated that adolescents living with HIV who were integrated into boarding schools or orphanages and had school staff members who were educated to support them, achieved better care and treatment outcomes [13,14].

The boarding school environment is an area where gains can be made to improve treatment outcomes for adolescents. In Kenya, there are about 4300 boarding secondary schools nationally, and of these, approximately 50% are public boarding schools [15,16]. According to the Kenya Population-based HIV Impact Assessment (KENPHIA) 2018 report, children were among those affected by the HIV burden, with about 139000 children aged 0–14 years living with HIV in Kenya [17]. We used qualitative methods to understand facilitators and barriers to school-based adherence support by exploring the school experiences of adolescents living with HIV and perceptions of caregivers and school nurses, to inform development of a supportive environment for antiretroviral adherence within boarding schools in Kenya.

## Methods

### Study design, setting, and participants

We used qualitative methods, including in-depth interviews (IDIs) with school nurses and focus group discussions (FGDs) with adolescents and caregivers, to explore experiences, perceptions, benefits, facilitators, and impediments to antiretroviral school-based support. We recruited adolescents and caregivers from Pumwani Hospital, and school nurses from boarding secondary schools within Nairobi County. Pumwani Hospital is one of the largest public facilities in Kenya serving a population of about 29000 and is located in a low-income urban area in Nairobi. The facility has an outpatient HIV clinic which supports HIV counseling and testing, prevention-of-mother-to-child-transmission, care and treatment, and virology laboratory services. The clinic serves about 370 adolescents living with HIV, about 80 of whom are enrolled in various boarding schools. Adolescent tailored services such as ART adherence, appointment keeping, sexual reproductive health, and mental health are offered on monthly basis in addition to adolescent day activities, universally referred as to Operation Triple Zero (OTZ) fun days, held quarterly [18]. The events focus on empowering adolescents and young people living with HIV to commit to "triple zero outcomes": zero missed appointments, zero

missed drugs, and zero viral load. The HIV clinic services are offered by clinical officers, nurses, counselors and peer mentors.

To get diverse views and increase applicability of our results, we purposively selected adolescents learning in boarding secondary schools with both male and female students. We included schools in national, county, and sub-county categories; and in urban, peri-urban, and rural settings. For each adolescent FGD, we recruited adolescents in each year of secondary school. Inclusion criteria for adolescents included ages 15–19 years, living with HIV, receiving HIV care and treatment at Pumwani hospital, and attending a boarding school. For caregiver FGDs, we recruited caregivers of adolescents in any level of boarding secondary school. Adolescents and caregivers were identified using the facility register and invited for a focus group discussion at the facility via a phone call by the HIV care nurses. For IDIs, we only recruited school nurses employed in Nairobi County boarding schools either in urban or peri-urban settings. The boarding schools were categorized into national, county, and sub-county levels. The boarding school selection was based on their location and proximity to the facility, and we prioritized closer boarding schools with the aim of recruiting at least 2 school nurses per level, having obtained prior school leadership permission. The inclusion criteria for school nurses included at least 1 year of boarding school experience and willingness to provide consent. There were no exclusion criteria.

## Data collection

Data was collected from July 2022 to November 2022. FGDs among caregivers and adolescents were conducted in-person at the health facility to explore views, benefits, barriers, experiences, and facilitators to the school-based adherence support; while IDIs among school nurses were conducted via phone. All discussions were audio recorded with participants' permission. The interview guides were developed by the investigators (NK, ACR, and KA) based on the topic of interest with additional inputs from a social scientist (EO). The guides were piloted among the caregivers and adolescents attending care and treatment in a different health facility prior to conducting the study. The FGDs were conducted by EO and EA, female social scientists, with Bachelor of Education and Bachelor of Arts degrees, and 2–5 years of experience in qualitative research methods, with the help of a research assistant (SK). IDIs were conducted remotely via phone depending on the school nurses' availability. The relative youthfulness of the facilitators was considered an advantage for working with adolescents during this project and both had prior experience in facilitating FGDs and IDIs. This project was funded by RHAY (Reducing HIV/AIDS in Youth) and the grant specifically funded youth researchers to get involved in HIV prevention. NK, ACR, and KA supported FGD and IDI guide development and provided supervisory oversight during data collection, coding, and analysis.

After explaining the purpose and procedure of the study to facility staff members, the research team were linked with interested caregivers and adolescents by HIV care facility nurses. Of note, researchers had no prior relationship with any of the participants. The facility staff line listed all adolescents living with HIV in boarding schools using facility registers and contacted their caregivers via a phone call. Only caregivers who were reached via phone call were given information on the purpose of the study, procedures, risks and benefits, and the scheduled FGDs dates. All caregivers were invited a week earlier and offered FGD attendance options based on their availability. The HIV care nurses had no influence towards adolescents participating in the FGDs, as FGDs were conducted during adolescent fun days at the facility. FGDs took place in a private room at the facility. Semi-structured topic guides were used in caregiver FGDs, focusing on their experience and views on school-based adherence; and in adolescent FGDs, focusing on their school life experience, acceptability, and views on school-

based adherence support. Both FGDs were conducted in either English or Swahili and were done by a social scientist, with help of a research assistant and each session lasted approximately two hours. After the first FGD sessions with caregivers and adolescents, the researchers further reviewed the guides and amended probes to reflect additional views during subsequent sessions. Note-taking was also done to capture the main ideas that arose as well as nonverbal information. The adolescent FGDs were conducted during adolescent day activities at the clinic during school holidays and invited adolescents were students in any boarding school in Kenya.

### School nurse in-depth-interviews

Using a semi-structured topic guide, in-depth interviews focused on the general school experience, barriers that could impede the success of in-person support of students living with HIV, and proposed strategies to implement school-based adherence support. School nurse interviews were conducted remotely via phone call by a social scientist, and each interview lasted about 40 minutes. Data collection was discontinued when thematic saturation was achieved, determined using the iterative approach.

### Data analysis

Focus group discussions and interviews were audio recorded, transcribed verbatim, translated into English where necessary, and anonymized using participant identification code and date. Data analysis was conducted using deductive and inductive thematic approaches and followed the steps described by Braun and Clarke [19], to facilitate finding of patterns and making meanings of data. Codebooks were developed iteratively, with initial codebooks developed based on FGD and IDI guides and revised using emerging themes from preliminary data. De-identified transcripts were imported into NVivo 12 (QSR International, Australia) [20] for coding. Ten percent of the transcripts were coded by two coders (NK and IM) to ensure inter-rater reliability and any differences were discussed until agreement was reached. The remaining transcripts were coded independently by NK and IM and integrated any additional emerging themes. Code excerpts were exported into Microsoft Word, and summaries written to capture the key themes from each code, focusing on topics related to facilitators and barriers to school-based adherence support from adolescent, caregiver, and school nurse perspectives.

### Ethical considerations

The study was approved by the Kenya Medical Research Institute (KEMRI) Scientific and Ethics Review Unit (Non-KEMRI 715). School nurses and caregivers signed informed consent forms and provided parental consent for minors under their care. Adolescents under 18 years also signed assent forms, while adolescents above 18 years signed informed consent forms.

## Results

We visited a total of 17 boarding secondary schools in Nairobi County: 3 boys' schools, 13 girls' schools, and 1 mixed (girls and boys) school with the aim of inviting nurses to participate in the IDIs. Four school administration and 3 school nurses declined to participate in the study, while 3 schools did not have a school nurse. Out of 65 caregivers line listed, 25 caregivers were reached via phone call and invited for FGDs, but only 11 turned up and no reason were sought for none attendance. For adolescent FGDs, about 114 children and adolescents attended fun day, and 18 met inclusion criteria i.e., above 18 years or having a parental consent. Most of the adolescents were familiar with each other and this could be attributed to their

active participation in fun day activities. The few adolescents who declined to participate in the FGDs were recent transfer-ins or newly diagnosed.There were no notable differences in girls and boys who declined to consent for participation; however a larger number of boys were in boarding schools compared to girls. Thus, majority of boys met our inclusion criteria and participated in the FGDs.

## Description of study participants

Overall, 36 individuals participated in this study, including 11 caregivers, 18 adolescents, and 7 nurses. Caregivers had a median age of 42 years, adolescents 17 years, and school nurses 32 years (**Table 1**).

**Table 1. Characteristics of participants.**

| Participant characteristics | | n (%) |
|---|---|---|
| **Adolescents living with HIV (n = 18)** | | |
| Gender | Male | 10 (56%) |
| | Female | 8 (44%) |
| Age | 15–16 | 4 (22%) |
| | 17–18 | 13 (72%) |
| | > 19 | 1 (6%) |
| Schooling level | Form 1 (Grade 9) | 3 (17%) |
| | Form 2(Grade 10) | 6 (33%) |
| | Form 3 (Grade 11) | 5 (28%) |
| | Form 4 (Grade 12) | 4 (22%) |
| **Caregivers (n = 11)** | | |
| Gender | Female | 11 (100%) |
| | Male | 0 |
| Age | 30–35 | 1 (9%) |
| | 36–40 | 4 (36%) |
| | 41–45 | 4 (36%) |
| | > 46 | 2 (18%) |
| Level of education | Primary level | 6 (55%) |
| | Secondary level | 4 (36%) |
| | Mid-level college | 1 (9%) |
| Occupation | Informal sector | 8 (73%) |
| | Formal sector | 3 (27%) |
| **School nurses (n = 7)** | | |
| Gender | Female | 6 (86%) |
| | Male | 1 (14%) |
| Level of education | Diploma | 7 (100%) |
| Age | 25–30 | 2 (29%) |
| | 31–35 | 3 (43%) |
| | 36–40 | 1 (14%) |
| | > 41 | 1 (14%) |
| Duration of school clinic work experience | 1 year | 1 (14%) |
| | 1–2 years | 2 (29%) |
| | 3–4 years | 3 (43%) |
| | > 5 years | 1 (14%) |

Table 1: Characteristics of Adolescents living with HIV, caregivers and school nurses interviewed on experiences and challenges of living with HIV in boarding schools n = 36 (Adolescents in boarding school n = 18, caregivers = 11 Providers n = 7).

### Adolescent HIV status disclosure in school settings

Some adolescents reported having disclosed their HIV status either to the school nurse, matron, teacher or a friend. Most of the adolescents' HIV status disclosure was facilitated by their caregiver to a school nurse during school admission:

> "*Before I was admitted [at boarding school], I went with my sister who disclosed my status to the nurse and so when I joined, she already had the information.*" (17-year-old female adolescent R4, FGD1)

> "*When I reported in form 1, we went to the deputy's office. I was there with my parent and the school nurse, and my parent told her about it and that I would be going to take the medication.*" (19-year-old female adolescent R9, FGD1)

Adolescents further reported having inadvertently disclosed their HIV status to their friends or teachers at school. This happened either when a friend saw the medication in their storage boxes, constantly being asked by a friend about frequent visits to the teacher or school nurse's office, or coincidentally meeting at the school nurse or teacher's office at the time of taking their ART:

> "*There was a day I was going to take my drugs and I found her at the teacher's office and the teacher had already placed the drugs on the table. So, she asked me if those really looked like sickle cell medication and I told her I didn't know. She then told me to tell her the truth that she wouldn't tell anyone, and I just told her.*" (18-year-old female adolescent R1, FGD2)

> "*My best friend had my keys at one time, and she went into my box and found the drugs. When she asked me, I just told her, and she has never told anyone.*' (17-year- old female adolescent R8, FGD2)

> "*She used to see me go to the matron every night and she kept on asking me why and if I was sick every day. I ignored her and then I just decided to tell her.*" (16-year-old female adolescent R4, FGD2)

The school nurses also reported having learned about the adolescents' HIV status either through reading school medical records or being informed by caregivers. The use of medical record forms at school was also reported by caregivers as means of HIV status disclosure:

> "*I told my daughter not to be scared to take her drugs. The school also knows as there is a form that you fill in to indicate. There was a time I went to take drugs to her I got to the gate and they called the office and when they looked at the file they confirmed that she is on medication so they just send her out to me without asking any questions.*" (45-year-old female caregiver R4, FGD2)

> "*All children who have disclosed to me were accompanied by their parents when they are being admitted in form 1. We have medical documents where they are required to indicate if they are suffering from a chronic condition. Thereafter they are directed to school nurse.*" (32-year-old female school nurse, IDI)

HIV status disclosure at school was further facilitated by the policy in some schools that does not allow students to store medication in their boxes at the dormitory or due to searching protocols at the school gate when students report to school. However, some school nurses, caregivers, and adolescents devised strategies to bypass this requirement while still ensuring the adolescents had their medication without other students or school staff finding out during the search:

"*We prepared a secret list of name students who should be exempted from search at the gate. Also, some of the caregivers do bring drugs for this adolescent the following day.*" (35-year-old female school nurse, IDI)

"*There are those who come with the parents when they are being admitted in form 1 because, in this school, every medication should be under my custody. Because no student should have medication, a parent will come and tell me the condition of the daughter and the reason for the medication.*" (27-year-old female school nurse, IDI)

Some adolescents bypassed search protocols by having their caregiver deliver drugs to them the following day or by being searched alone at the gate and stating that the drug is for another condition.

"*We usually talk to them and make sure that they also carry their medication when they are reporting to school. With the drugs, I will carry them and give them to the nurse because if they carry them in their bags they will be searched, and everyone will find out that they are using those drugs.*" (39-year-old female caregiver R4, FGD2)

''*I am checked with whomever I find at the gate alone and when they ask me about the drugs, I tell them that they are for my chest.*'' (19-year-old female adolescent R2, FGD2)

## Approaches to HIV status disclosure in a boarding school setting

Caregivers reported discussing with their adolescents about HIV disclosure at school and sought their permission prior to reporting to the boarding school. The caregivers further expressed a need to be involved in facilitating adolescents' HIV status disclosure to the school nurse. Some caregivers, adolescents, and school nurses reported barriers to HIV disclosure at school, including adolescents declining to disclose their HIV status, lack of confidentiality within the school, adolescents not being aware of their HIV status, and fear of discrimination from fellow students:

"*For me, I think it important to inform them [nurses] of the status of the child so that she can know how to handle her. I will do it as the child can't do it by themselves but before that, I will sit down and talk to my daughter first, then all of us together with the nurse can have a sit down.*" (42-year-old female caregiver R1, FGD2)

"*The child should be willing to disclose the status. You might want to take her to the nurse, and she is not willing. She has to be willing.*" (36 year-old-female caregiver R6, FGD1)

"*I fear to disclose because I fear the society. What will they think about me, will they start looking at me differently, yet I want to live a normal life.*" (17-year-old female adolescent R8, FGD2)

## Boarding school experience of adolescents living with HIV

Adolescents shared their lived experience in boarding schools regarding how they take their antiretroviral treatment. They noted challenges within the school environment, medication

storage, school nurse interaction, coping strategies, and conformability to discuss topics on HIV/AIDS among peers. Other challenges reported by the caregivers and school nurses included lack of privacy and confidentiality, stress, HIV seropositive status denial, and stigma:

> "*The challenge that I have is that when keeping drugs for yourself and so you are afraid of your friends getting into your box, seeing the drugs and then starting to ask questions.*" (17-year-old male adolescent R8, FGD2)

> "*Acceptance is an issue especially when she comes to a new environment and maybe in her class, she feels she is the only one. So, if she feels that way, she may want to start missing taking the medication.*" (33-year-old female school nurse, IDI)

> "*Here in school, we have a problem with stigma issues and poor adherence. For example, some of the students have transferred to other schools after the other student knew their HIV status. This happened during dormitory search and students were able to see the drugs.*" (48-year-old female school nurse, IDI)

> "*When I was in form 1, I told my friend called [A] who later transferred. I trusted her and told her that I was HIV positive and after 1 week, I started hearing rumors about it. I cried a lot and went to the deputy. The deputy talked to her and even gave her some punishment*". (17-year-old female adolescent R4, FGD)

> "*For example, with these guidance and counselling teachers, as soon as you leave that room, you become the topic of discussion. What you disclose to that teacher, she will disclose to other teachers and so in class, there is a way the teacher looks at you. These teachers like backbiting people*" (18-year-old female adolescent R7, FGD)

## Coping strategies in school settings

Adolescents reported different strategies for taking and storing their antiretroviral medication and seeking support in school. Some adolescents reported having considered taking their drugs when other students are engaged with other school routine activities, waking up earlier, excusing themselves, and disclosing that their drugs are for different diseases e.g., ulcers:

> "*You find it difficult to take drugs when your friends are there because they keep on asking you why you are taking drugs. So, I used to pretend that I have fainted so that I am taken to the deputy, and then I can take drugs from there.*" (17-year-old female adolescent R7, FGD2)

> "*I take mine in the morning only and I lie that it is an allergy medication. I also changed the storage bottle; my mum changed it for me so even if they see the bottle, they will just think it is medication for my chest.*" (18-year-old female adolescent R3, FGD1)

In addition, adolescents who stored their medication with school nurses had an individualized specific time to take their medication at the nurse's office or were issued pills for a specified number of days. In some schools, these adolescents have been allocated lockable cabinets or bags at the school clinic which they could access at their own convenient time to take the medication:

> "*. . .. We talked, and she told me that for such cases the child is given a key to the room where she will be able to find water and drugs labeled under her name. When it is time for her to take the medicine, no one will even know where she has gone. She has the keys and can access her drugs personally.*" (40-year-old female caregiver R1, FGD2)

"*For illnesses like asthma or ulcers, we keep the medication in another cupboard and label it with a name. With HIV, everyone has a small bag to help with separating the medicine so that when they come, everyone picks theirs.*" (27-year-old female school nurse, IDI)

"*When these students come to the clinic, of course, they will find a lot of students there, but I like giving them the first priority. In my clinic, there is a waiting area and a consultation room. So, for the consultation, all students get into the consultation room, and I close the door.*" (35-year-old male school nurse, IDI)

## Peer-to-peer discussions about HIV and sex at school

Some of the adolescents reported shying away from contributing to topical issues around HIV/AIDS due to fear of being perceived to know too much concerning the topic, which might lead to others suspecting that they are living with HIV:

"*I would rather pretend that I don't know than join in the discussion because they will start looking at me like I know too much information.*" (16-year-old male adolescent R2, FGD-1)

"*I am not comfortable because sometimes they speak ill about the people living with HIV and you want to correct them, but you can't because they will think that you have it.*" (17-year-old female adolescent R9, FGD1)

"*I feel comfortable because I pretend that I am like them.*" (18-year-old female adolescent R8, FGD1)

However, other adolescents were comfortable discussing HIV and sex issues, and some used the opportunity to debunk myths of HIV. This was because their friends were not aware of their HIV status. Both negative and positive issues were discussed among these peers regarding people living with HIV:

"*I am comfortable because sometimes they ask me what I would do if my best friend has it and I tell them that I will accept her because there is no need for me to make her lose hope. That we need to make them feel loved and that they have sisters around them.*" (17-year-old female adolescent R7, FGD2)

"*Sometimes I hear them say that if they get their friends or boyfriends [HIV-] positive, they will end the relationship: because I can't tell them my status, I just tell them that people don't die from it. Once you accept yourself and start taking drugs, you become healthy.*" (18-year-old female adolescent R6, FGD2)

## Adolescent-school nurse interaction

Adolescents reported having minimal interaction with school nurses when taking their drugs at the school clinic. This was because adolescents saw no need to engage further with the school nurse as they were adhering to their drugs well:

"*First of all, we had an agreement that she wouldn't ask me those questions and then I just go and refill my medication after every 5 days. We don't talk, I know where the drugs are and so I get in, take them, and leave.*" (18-year-old male adolescent R4, FGD1)

School nurses had a different perspective on adolescents' minimal interaction with them, suggesting that this could be because adolescents don't like talking about their issues, or that

they feared their peers would notice them spending time with the nurse for guidance and counseling sessions:

> "*You see, for most of these girls, because of their age, they are not able to open up: they are very shy. So, until you as a grown-up sense that this student is not the same way that I am used to her, you cannot know that she has a problem. Very few girls will come out to share their issues.*" (39-year-old female school nurse, IDI)

> "*There are those girls that fear going for counseling sessions because they know that, that room is for guidance and counseling and so if people see me getting in there, they will know that I have an issue.*" (32-year-old female school nurse, IDI)

## Adherence reminders and reasons for skipping medication

Adolescents reported various techniques they used to remember to take medication. These included taking their medication when doing routine activities like brushing their teeth, relying on their instincts, and by use of watch alarms. Others were reminded by friends or school nurses to take their medication. The school nurses were reported to remind adolescents to take their medication when they became aware of missed doses through pill counts, or noticed that an adolescent had not come for medication on a particular day:

> "*When we come from the parade (assembly), I just remember or sometimes my friend just asks me why I'm not going to the sick bay.* "(16-year-old male adolescent R2, FGD2)

> "*We go around and check on whether these students are taking their medicine. We look at the numbers by counting [pill count].*"(30-year-old female school nurse, IDI)

The adolescents reported that sometimes they skipped taking their medication either due to school activities like attending games, going to church, finding many students at the clinic, or just forgetting:

> "*I skip when there are many students at the nurse's office because it will draw attention to me.*" (17-year-old female adolescent R6, FGD1)

> "*So, when we are supposed to take drugs, you find other students in the cube who have come to tell stories and so we join them and forget.*" (15-year-old male adolescent R4, FGD2)

## Peer-to-peer support in boarding school

School nurses and adolescents reported that some student had at least one peer supporter. Peer support was either established by the students themselves after they discovered they were mutually living with HIV; or was initiated by either peers or the school nurse. In addition, the caregivers also reported that some adolescents knew each other's HIV status and encouraged one another:

> "*There are 2 of us who are [HIV-] positive, and we keep our drugs together. For example, when we are about to take drugs, she pretends that there is something she is taking from the box, she gets the drugs, and she calls me. I in turn come with the water and we take the drugs.*" (17-year-old female adolescent R4, FGD2)

> "*When I took mine (daughter) to the school, the nurse knew, and she was in the same situation as her. That nurse is now gone but she met another girl like her, and they normally talk to*

*each other. I have even come across messages of them encouraging each other. They met at the nurse's office.*" (45-year-old female caregiver R2, FGD1)

"*There was this very outspoken girl, and she was proud of who she was. So, this outspoken girl was the one who started it [peer-support]. So, I was now able to take it up even after she had left. She could even visit the form 1 classes after admission and introduce herself and tell them that she is the ambassador of that [HIV advocate] and tell them to feel free in case they wanted to talk. Through that, she was now able to bring them (adolescents) to me.*" (32-year-old female school nurse, IDI)

## Adolescent opinions about school nurses

Adolescents shared what they liked and disliked about their school nurses, with most reporting that school nurses were friendly, understanding, fair, and treated them well and confidentially. Caregivers also reported that school nurses provided support; and importantly, a loop of communication back to the caregivers:

''*She is friendly, and she isn't someone who will go about disclosing your status.*" (17-year-old female adolescent R7, FGD2)

''*The nurse is okay. My child's nurse even calls to find out when is the next clinic date for my child. Sometimes over the weekend she will call her and talk to her to see how she is faring. Then she will later call me to update me.*" (34-year-old female caregiver R4, FGD2)

## Views and recommendation on school-based adherence support

Caregivers and adolescents reported varied views regarding school-based antiretroviral adherence support. Some concerns they raised included nurses being too strict and unsympathetic to adolescents, fear of unnecessary attention, and lack of confidentiality.

''*There are some nurses that are so strict with the children and don't become friends with them and even end up embarrassing the child. They sometimes neglect other children and don't do enough follow-up, there was also an incident of the child passing away. So, my child told me that it is better for her to just take the medication by herself.*" (46-year-old female caregiver R1, FGD1)

Some of the caregivers were not in support of school-based enhanced support sessions due to fear of unnecessary attention that could arise if the model is implemented, as it can lead to inadvertent disclosure of HIV status. Other caregivers were in support of general counseling talks being done to all students regardless of their HIV status, in order to minimize attention to students living with HIV:

''*I don't agree with that [school-based enhanced support]. When you call only the children involved, the other children in the school will ask questions about why it is only a few students being counseled.*" (42-year-old female caregiver R2, FGD1)

''*If they can organize a time to talk to all the children in general about these things [HIV]. But the targeted children will know that the message is of more use to them.*" (40-year-old female caregiver R1, FGD2)

Adolescents reported having regular guidance and counseling sessions facilitated by their teachers, that covered topics on self-esteem, alcohol, and drugs. Some adolescents reported

being comfortable with school-based adherence support sessions, while others preferred to wait for facility-based counseling due to likelihood of unwarranted attention. In addition, some adolescents reported talking to class teachers or school nurses whom they had disclosed their HIV status to, when they felt stressed, while others did not:

> "*When called upon, I will go because there are people who have not accepted themselves, and maybe if they see others, they will accept comfortably.*" (18-year-old male adolescent R8, FGD2)

> "*The guidance and counseling teacher talks to the whole class at once and what is said is normally repeated all the time. It is okay just the way it is. I would rather wait for 3 months to get the counseling here (at facility) than get it in school.*" (16-year-old male adolescent R2, FGD1)

> "*I don't trust these teachers. When I am stressed, I go to the deputy who knows my status, call my mother and talk it out.*" (17-year-old female adolescent R7, FGD2)

Caregivers, adolescents, and school nurses recommended strategies to be put in place if school-based adherence support was to be offered for those who are comfortable with it. Caregivers expressed the need for all boarding schools to have a trained nurse to support adolescents in storing their medication and guiding them on pill-taking, and a need for constant communication between caregivers and school nurses on adolescents' progress at school. School nurses also highlighted need for necessary training:

> "*The matron and the school nurse could go for a workshop and be trained about these issues; it could really help. This is because there are those students who will go to a matron and talk to them but maybe the matron or guidance and counseling teacher does not have the skills to approach the issue. And so, the student will be left wondering why they should take the medication.*" (39-year-old female school nurse, IDI)

## Discussion

We explored adolescent life experiences in Kenyan boarding secondary schools and sought school nurse and caregiver views on school-based ART adherence support. Our findings show that schools have potential for; and are already supporting adolescents living with HIV with storing their medication and providing adherence support through teachers, peers, and school nurses. From our findings, adolescent willingness to disclose HIV status, caregiver facilitation, and school nurse skills and experience, are key drivers for school-based adherence support. These findings also demonstrate a need for caregiver-initiated early preparation for adolescent antiretroviral adherence support transition to boarding school environments, and the establishment of strong school nurse-adolescent rapport from the outset.

Other researchers have found that HIV-related stigma compromised clients' ability to successfully adhere to ART [21–23]. In our study, pervasive HIV-related stigma was a key consideration for adherence, as adolescents living with HIV, their caregivers, and school nurses all described multiple ways that they worked together to keep HIV medication a secret from other people. These adolescents experience internalized shame and fear for years at school which could have long-term consequences to their quality of life and wellbeing. While all respondents recognized that additional support in the form of nurses and counseling could be helpful, fear of stigma continued to be the main motivator for adolescents living with HIV and their caregivers when considering options at boarding schools.

Despite the stigma challenge, caregivers and adolescents devised numerous strategies to achieve both goals of adhering to medication and preserving the secrecy of the adolescents' HIV serostatus. We found that caregivers are deliberately seeking school-based support through facilitating adolescents' HIV status disclosure and delivering medication, while schools are using medical records to identify students in need of medical support and linking them to school nurses. These findings are in agreement with other studies which conclude that families are the main source of caregiving for ALHIV [24], and that they should not be left to shoulder the burden of caring for ALHIV alone [25]. Despite the foregoing support, students and caregivers did not prefer routine school-based adherence support sessions due to the risk of drawing attention of other students. These findings are similar to other studies which showed that patients feared taking their medication in the presence of other people as it would lead to suspicion or inadvertent disclosure of their HIV status [26].

Most adolescents regarded school nurses as friendly and empathetic, which may create opportunities for tailored school-based adherence interventions. In addition, adolescents already talk to teachers and nurses when they feel stressed, while caregivers view nurses as an important loop of communication on adolescents' progress. These findings agree with those from other studies which showed that social protection mechanisms facilitated reach to adolescents living with HIV in need, either through food support, school allowance, transport allowance to ART clinics, psychosocial counseling, vocational training, or home visits [27]. However, other researchers have recognized that a holistically supportive school environment is hard to achieve. According to Kimera et al. [28], building HIV/AIDS-care and support-competent schools is a lengthy, deliberate process that calls for each school to have a solid understanding of HIV/AIDS, support requirements, and stakeholders' participation in the design of a holistic strategy. The adolescents we interviewed were not in environments that met this standard, and the nurses did not have the requisite training to support the adolescents beyond occasional pill-taking reminders. This finding correlates with another study which found out that there was meager support for HIV prevention available in schools due to structural factors [29].

HIV disclosure is typically determined by the level of support expected. Factors that influence HIV disclosure within a family may include lack of knowledge, fear of stigmatization, social rejection, and discouragement from friends and family [30]. We found similar factors related to school-based HIV disclosure, including adolescents' willingness to disclose their HIV status to people at school, fear of discrimination, and perceived lack of confidentiality in a school setting. All of these were barriers to HIV status disclosure to both school nurses and teachers; and are in agreement with other findings on how complex challenges surrounding HIV status disclosure contribute to ALHIV disengagement from care [31,32]. Other researchers noted similar factors, with one study stating that adolescent boys and young men living with HIV kept their condition a secret to fit in with their peers, and to shield themselves from discrimination and loneliness [33]. However, in our findings there were no notable differences in the opinions among male and female adolescents.

According to Shona et al., asymptomatic PLHIV were motivated to initiate ART in order to conceal stigmatizing symptoms and were continually assessing risks and navigating potential exposure while engaging with treatment and care services [34]. The adolescents in our study were resourceful as they often innovated despite the stigma they experienced, describing many unique coping strategies to successfully adhere to their medication and keep their serostatus a secret while at school. These included opting for HIV non-disclosure, lying about their medication and health condition, changing medication storage containers, moderating their participation in peer discussions on HIV issues, and negotiating for priority services in their school

clinics. These finding are similar to those of Skemitt et al., which explored how factors and priorities influenced satisfaction with care [35].

The adolescents we spoke to reported that their individual systems were working sufficiently to keep them on their medication, and they appreciated the support they were receiving in school. Both caregivers and adolescents expressed the need for all boarding schools to have a trained nurse who understood the adolescents' special needs while at school, to appropriately support, guide, and advise them. According to Toromo et al., there is a need to enhance training and resources for health care workers to support adolescents during and after HIV status disclosure [31]. However, nurses indicated that they did not have the resources needed to optimally serve this population.

Peer models have been used successfully in other settings; in the Zvandiri program model in Zimbabwe, Community Adolescent Treatment Supporters (CATS) were able to deliver adherence and psychosocial support in their communities after they were trained [36]. Therefore, there is an urgent need to train school nurses on antiretroviral adherence counseling and psychosocial support; and equip them with skills for adherence advocacy within boarding school settings. The root causes for the reluctance of adolescents to engage with school nurses were unclear. However some pointed out that school nurses lacked skills and knowledge on how to build a good rapport with adolescents and these findings were in agreement with those of Edith et al., on barriers to ART adherence in school settings [37].

In a recent systematic review, individualized care plans, communication, psychological support, and health and sexual education were considered high-quality evidence transition interventions [38]. During our discussions with respondents, we discovered a need to strengthen the provision of counseling on transition to boarding schools. Caregivers spoke of extensive preparations taken to successfully prepare the school environment to support their adolescents in secondary school. Our discussions showed that successful transition included caregivers engaging adolescents in the process, including obtaining their assent to disclose their HIV status to school leadership, and caregivers actively assuming the role of transition-facilitators, by linking adolescents to school nurses. This process parallels some of the work already being done in Africa to support youth living with HIV as they transition to adult HIV care, where best practices include equipping adolescents with relevant information about their condition, self-advocacy skills, and support from caregivers [39,40].

## Limitations

Our sample may not be representative of all populations for multiple reasons. First, adolescents and caregivers may have been reluctant to participate in the study due to unwillingness to disclose their HIV status to the researchers or other participants in an FGD. We may also have recruited a population that is more comfortable speaking openly about their HIV status. Additionally, adolescents enrolled in boarding secondary schools have potentially already navigated many educational and financial barriers and may be more privileged than average adolescents living with HIV. Further, FGDs were in mixed-gender groups which may have influenced the type of information disclosed and shared by participants. We also only interviewed school nurses from the Nairobi area, hence their experiences may have been different from those in rural schools; and nurses who were more interested in, or more accepting of HIV may have been more likely to participate in interviews. However, our study had the strength of drawing on individual experiences on seeking school-based support across a continuum of ART adherence. In addition, researchers participating in this study were from different backgrounds e.g. nurse, medical doctor and social scientist, and all acknowledge their bias, own beliefs and judgement systems going into this analysis. Reassuringly, having investigators from mixed

backgrounds helped to minimize any bias that would have been driven by a single individual's background. Despite a lack of ethical approval to interview teachers, this study has revealed that teachers are also important and any study following on from this one will include teachers.

## Conclusion

Adolescents described operating in highly stigmatized school environments in which maintaining secrecy about their health, HIV serostatus, and medication was a fact of daily life, hence they have devised strategies to navigate pill-taking while enlisting quiet support. Caregivers described actively managing the transition to boarding secondary schools, which helped link adolescents to supportive adults to allow successful medication-taking. In schools, positive peer and adult role models facilitated adherence to medication, while stigma, lack of adult or peer support, and fear of inadvertent disclosure hindered adherence in some environments. It is evident that adolescents living with HIV would benefit from a less stigmatizing school environment. Increasing school nurses' expertise in supporting adolescent HIV treatment adherence, and the involvement of adolescents in future interventions design to promote HIV medication adherence, could improve outcomes for adolescents living with HIV in boarding schools.

## Supporting information

**S1 Checklist. COREQ (Consolidated criteria for REporting Qualitative research) checklist.**
(PDF)

**S1 Codebook. Adolescent FGD.**
(PDF)

**S2 Codebook. Caregiver FGD.**
(PDF)

**S3 Codebook. School nurse ID.**
(PDF)

**S1 Text. FGD Caregiver guide.**
(PDF)

**S2 Text. FGD Adolescent guide.**
(PDF)

**S3 Text. IDI school nurse guide.**
(PDF)

## Acknowledgments

We acknowledge the contributions of all caregivers, adolescents, and school nurses. We thank Eunice Adhiambo, Susan Kathodheki, and facility staff for assisting with data collection and the RHAY secretariat team for administrative support.

## Author Contributions

**Conceptualization:** Nicholas Kipkurui, James Ayieko, Kawango Agot, Alison C. Roxby.

**Data curation:** Nicholas Kipkurui, Emmah Owidi, Irene Mugenya.

**Formal analysis:** Nicholas Kipkurui, Irene Mugenya, Alison C. Roxby.

**Funding acquisition:** Kawango Agot.

**Investigation:** Emmah Owidi, Alison C. Roxby.

**Methodology:** Nicholas Kipkurui, Emmah Owidi, James Ayieko, Kawango Agot, Alison C. Roxby.

**Resources:** Gerald Owuor.

**Supervision:** Nicholas Kipkurui, James Ayieko, Gerald Owuor, Kawango Agot, Alison C. Roxby.

**Validation:** Gerald Owuor, Irene Mugenya, Kawango Agot.

**Visualization:** Gerald Owuor.

**Writing – original draft:** Nicholas Kipkurui, Kawango Agot, Alison C. Roxby.

**Writing – review & editing:** Nicholas Kipkurui, Emmah Owidi, James Ayieko, Gerald Owuor, Irene Mugenya, Kawango Agot, Alison C. Roxby.

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
