## [Decision Letter · Decision Letter 0]

16 May 2023

PGPH-D-23-00355

How adolescents living with HIV navigate antiretroviral adherence in boarding schools in Nairobi, Kenya: a qualitative study of adolescents, caregivers and school nurses

Dear Dr. Kipkurui,

Thank you for submitting your manuscript to PLOS Global Public Health. After careful consideration, we feel that it has merit but does not fully meet PLOS Global Public Health’s publication criteria as it currently stands. Therefore, we invite you to submit a revised version of the manuscript that addresses the points raised during the review process.

EDITOR comments:

My appreciation to the authors for this insightful work. Please address the feedback from reviewers 1 and 2 and resubmit the manuscript for further consideration. A few comments from myself for your consideration:

1. Re: data analysis: "Analysis followed deductive and inductive thematic approaches, with initial categories based on the interview guide coded by NK and emergent themes integrated into the second round of coding by IM." Please provide some more expansive details on qualitative data analysis-was there a theory or model used for analysis? what type of deductive and thematic approach(es) was/were used? Cite any sources applicable. There was no clear indication (per requirements of the COREQ checklist) of methodological orientation underpinning the study.

2. Related to the above, the pages cited in the COREQ checklist do not appear to correlate with the main manuscript. I take it those are line numbers and not page numbers, if so, please indicate in the COREQ checklist. Also correct the first "page number" listed as "2rd" for interviewer/facilitator. 

3. Section titled "Adolescents Boarding School Experience" should really indicate that this was adolescents *living with HIV*, otherwise it reads like a general description of adolescents in boarding school. Suggest "Boarding School Experience for Adolescents Living with HIV." Same comment for "Adolescent" section in Table 1. It should be ALHIV (expand the term if not defined as a table footnote).

4. Discussion, line 367: Other researchers *have* (not HAS) found....Please review th emanscript thoroughly for grammar and syntax errors. It may be good to have an author or non-author with strong English editing skills to review so that none of these errors are missed. 

We look forward to receiving your revised manuscript.

Kind regards,

Nadia Adjoa Sam-Agudu, M.D.

Academic Editor

Journal Requirements:

2. We have noticed that you have uploaded Supporting Information files, but you have not included a list of legends. Please add a full list of legends for your Supporting Information files after the references list. 

3. In the online submission form, you indicated that "De-identified participant data, including interview transcripts, will be made freely available after publication upon request to the corresponding author and with concurrence of the KEMRI Scientific and Ethics Review Unit". All PLOS journals now require all data underlying the findings described in their manuscript to be freely available to other researchers, either 1. In a public repository, 2. Within the manuscript itself, or 3. Uploaded as supplementary information.

Additional Editor Comments (if provided):

Reviewers' comments:

Reviewer's Responses to Questions

**Comments to the Author**

1. Does this manuscript meet PLOS Global Public Health’s publication criteria? Is the manuscript technically sound, and do the data support the conclusions? The manuscript must describe methodologically and ethically rigorous research with conclusions that are appropriately drawn based on the data presented.

Reviewer #1: Yes

Reviewer #2: Yes

2. Has the statistical analysis been performed appropriately and rigorously?

Reviewer #1: I don't know

Reviewer #2: N/A

3. Have the authors made all data underlying the findings in their manuscript fully available (please refer to the Data Availability Statement at the start of the manuscript PDF file)?

Reviewer #1: Yes

Reviewer #2: Yes

4. Is the manuscript presented in an intelligible fashion and written in standard English?

Reviewer #1: Yes

Reviewer #2: Yes

5. Review Comments to the Author

Reviewer #1: Manuscript Review Comments

Overall, a good, informative and commendable work relevant to exploring and addressing the needs of ALHIV.

LINE 2-3: Consider revising this statement to refer to boarding secondary schools rather than secondary schools. This is because being a student in day secondary schools may not have a directly affect ART adherence.

LINE 14: The word some ALHIV does not give readers an idea of the proportion who disclosed. Perhaps rewording to reflect a rough estimate using words such as few, most, majority etc.

LINE 47-48: It would enrich the paper to include information indicating the estimated number/proportion of ALHIV attending boarding schools if such data is available. This would showcase the magnitude of the problem and potential impact of the suggested interventions among ALHIV.

LINE 62: If there was more than one nurse per level, how was recruitment for participation in IDI done? Provide more information on the participants’ selection process.

LINE 63-64: Any exclusion criteria? If there was, it is important to mention.

LINE 71-72: The study setting is not clearly described. There is need to provide overview of the demographic, psychosocial and other relevant characteristics of the settings, especially as it relates to potential stigma and discrimination of adolescents living with HIV, procedures for access to available HIV screening, treatment and counselling care services, cadre and level of experience and relevant training exposure of healthcare workers at the facility and schools.

LINE 81-83: Does this statement imply that EO and EA were undergraduates and female scientists? ‘EO and EA, BA undergraduate, female social scientists with 2-5 years of experience in qualitative research facilitated FGDs among caregivers and adolescents with help of a research assistant’. Considering that the entire data collection for FGD and IDI were managed by undergraduates despite their 2-5 years of experience, was there any supervisory oversight by potentially more experienced authors NK, ACR and KA, especially to ensure that developed guides were utilized appropriately as designed?

LINE 85-86: In view of sensitivity of the topic, did the facility staff members use any criteria to identify interested caregivers and adolescents, especially while maintaining privacy, confidentiality and voluntary participation? For instance, did the facility staff have prior knowledge of the adolescents’ personalities and potential disposition towards participating in FGDs? It would be good to see how many did not consent, as well as rationale for non-consenting. Lessons may be useful for future research/researchers in similar settings.

LINE 114-120: Though there was disproportionately higher number of girls-only compared with boys-only schools (13 vs 3), there were more male than female participants (10 vs 8). The rationale for this would be good to know. Was it that more girls than boys declined consent to participate? This distribution is not clear, considering that 13 consenting girl-only schools were visited and there were only 8 female participants presented here. Kindly clarify. Also, in table 1, thought the numbers are small, it would still be good to include the proportions/percentages alongside the numbers in the last column. This will aid comparison across subgroups.

LINE 120: Not all potential audience (settings) would understand Form in the context of the ALHIV's class as it may not be a universally known terminology. Consider adding in bracket alternate meaning of form e.g grade equivalent.

LINE 163-165; 170-173: These lines provide commendable attempt at balance of expression of strategies for (maintaining) non-disclosure of HIV status. However, these are strategies utilized by the school nurse/management only. It would be good to see expressions of non-disclosure strategies employed by the adolescents, especially while they live and interact with their peers. Also, if it is extractable from the interviews, it would be good to see verbal and non-verbal cues that suggest degree of confidentiality, privacy, peer interaction and other relevant perceived psychosocial effects of HIV status disclosure as well as non-disclosure especially among the adolescent participants.

LINE 190: Consider qualifying this statement in the context of antiretroviral treatment. “Adolescents’ boarding school experience” may be too broad.

LINE 406-408: Was there a difference in opinions and/or strategies between female and male ALHIV? If there was consider reporting these in your result/discussion.

LINE 457: This statement may be true but in the result section, no quotes were mentioned to showcase the forms of stigma experienced/reported by the ALHIV participants. Most of the quotes pointed to the fear of stigma rather than stigma experienced. Consider bringing out quotes that showcase experiences of stigma if there were.

Reviewer #2: I will like to thank the author for a well-written qualitative study into a critical mass of ALHIV i.e those navigating the boarding school environment.

The author may kindly respond to the following concerns:

Major issues

1. There is no citation in text for the reference number 3 which goes to explain why references for Kimera et al and Skerritt et al are one reference away from those cited in text.

2. Secondary reference for global ALHIV data in a paper from Togo; certainly there are UNAIDs/WHO data that can be referenced.

3. Though references made to teachers being facilitators/barriers to adherence etc e.g. in lines 361 and 387, any reason why their views were not sought?

Minor issues

1. Line 20- harmonization of terms. "adolescents" rather than "teens"

2. Line 276 - ...why am [sic].....

3. Line 288- "...students" instead of "...student"

4. Line 313- "...faring" instead of "..fairing"

5. Line 383- "..another study" instead of "other study"

6. PLOS authors have the option to publish the peer review history of their article (what does this mean?). If published, this will include your full peer review and any attached files.

**Do you want your identity to be public for this peer review?** For information about this choice, including consent withdrawal, please see our Privacy Policy.

Reviewer #1: **Yes: **Tongdiyen Laura Jasper

Reviewer #2: **Yes: **Charles Martyn-Dickens

---

## [Decision Letter · Decision Letter 1]

4 Aug 2023

PGPH-D-23-00355R1

Navigating antiretroviral adherence in boarding secondary schools in Nairobi, Kenya: a qualitative study of adolescents living with HIV, their caregivers and school nurses

Dear Dr. Kipkurui,

Thank you for submitting your manuscript to PLOS Global Public Health. After careful consideration, we feel that it has merit but does not fully meet PLOS Global Public Health’s publication criteria as it currently stands. Therefore, we invite you to submit a revised version of the manuscript that addresses the points raised during the review process.

We look forward to receiving your revised manuscript.

Kind regards,

Miquel Vall-llosera Camps

Staff Editor

Journal Requirements:

Reviewers' comments:

Reviewer's Responses to Questions

**Comments to the Author**

1. If the authors have adequately addressed your comments raised in a previous round of review and you feel that this manuscript is now acceptable for publication, you may indicate that here to bypass the “Comments to the Author” section, enter your conflict of interest statement in the “Confidential to Editor” section, and submit your "Accept" recommendation.

Reviewer #1: All comments have been addressed

Reviewer #2: All comments have been addressed

2. Does this manuscript meet PLOS Global Public Health’s publication criteria? Is the manuscript technically sound, and do the data support the conclusions? The manuscript must describe methodologically and ethically rigorous research with conclusions that are appropriately drawn based on the data presented.

Reviewer #1: Yes

Reviewer #2: Yes

3. Has the statistical analysis been performed appropriately and rigorously?

Reviewer #1: Yes

Reviewer #2: I don't know

4. Have the authors made all data underlying the findings in their manuscript fully available (please refer to the Data Availability Statement at the start of the manuscript PDF file)?

Reviewer #1: Yes

Reviewer #2: Yes

5. Is the manuscript presented in an intelligible fashion and written in standard English?

Reviewer #1: Yes

Reviewer #2: Yes

6. Review Comments to the Author

Reviewer #1: (No Response)

Reviewer #2: I thank the authors for their kind responses to our comments raised

.

Kindly find some minor comments and corrections required prior to publication:

Could the authors give any comment as to the source of HIV in these clients. A pervasive assumption is that they acquired HIV vertically. if so were the concerns of Mothers in FGDs also a revelation of concern that their status were to be disclosed with school based interventions?

Line 16 " school nurse friendliness, being understanding, fair and confidential" not '..fairness, '

Line 51 should read approximately 50% not approximate 50%

Line 115-116- can you clarify who considered the youthful age of social science investigators an advantage? was it the participants or the research team?

Line 136 - is the term "learner" of particular significance? if not can these adolescents be referred to as "students"?

Line 143- does data saturation refer to thematic saturation?

line 170- was any attempt made to find out why 14 of 25 invited guests did not show up?

Line 323- I believe the sentence should be .."debunk (myths) of HIV"

Line 519.... these findings (plural form)

Line 545.... reword the sentence, "despite a lack of approval to interview teachers,......."

Thank you

7. PLOS authors have the option to publish the peer review history of their article (what does this mean?). If published, this will include your full peer review and any attached files.

**Do you want your identity to be public for this peer review?** For information about this choice, including consent withdrawal, please see our Privacy Policy.

Reviewer #1: No

Reviewer #2: **Yes: **Charles Martyn-Dickens

---

## [Decision Letter · Decision Letter 2]

23 Aug 2023

PGPH-D-23-00355R2

Navigating antiretroviral adherence in boarding secondary schools in Nairobi, Kenya: a qualitative study of adolescents living with HIV, their caregivers and school nurses

Dear Dr. Kipkurui,

Thank you for submitting your manuscript to PLOS Global Public Health. After careful consideration, we feel that it has merit but does not fully meet PLOS Global Public Health’s publication criteria as it currently stands. Therefore, we invite you to submit a revised version of the manuscript that addresses the points raised during the review process.

EDITORS COMMENTS: Kindly add a brief note on Reflexivity (researchers perspective) as well as trustworthiness in the Limitations- which are better titled 'Methodological Considerations'

Kindly address the comments from the reviewers.

We look forward to receiving your revised manuscript.

Kind regards,

Rashmi Josephine Rodrigues, M.D., Ph.D.

Academic Editor

Journal Requirements:

Additional Editor Comments (if provided):

Reviewers' comments:

Reviewer's Responses to Questions

**Comments to the Author**

1. If the authors have adequately addressed your comments raised in a previous round of review and you feel that this manuscript is now acceptable for publication, you may indicate that here to bypass the “Comments to the Author” section, enter your conflict of interest statement in the “Confidential to Editor” section, and submit your "Accept" recommendation.

Reviewer #1: All comments have been addressed

Reviewer #2: All comments have been addressed

2. Does this manuscript meet PLOS Global Public Health’s publication criteria? Is the manuscript technically sound, and do the data support the conclusions? The manuscript must describe methodologically and ethically rigorous research with conclusions that are appropriately drawn based on the data presented.

Reviewer #1: Yes

Reviewer #2: Yes

3. Has the statistical analysis been performed appropriately and rigorously?

Reviewer #1: Yes

Reviewer #2: I don't know

4. Have the authors made all data underlying the findings in their manuscript fully available (please refer to the Data Availability Statement at the start of the manuscript PDF file)?

Reviewer #1: Yes

Reviewer #2: Yes

5. Is the manuscript presented in an intelligible fashion and written in standard English?

Reviewer #1: Yes

Reviewer #2: Yes

6. Review Comments to the Author

Reviewer #1: Thanks to the authors for addressing all previous comments.

Please see below a few comments for correction prior to publication.

Line 16 read better as: “School nurse friendliness, being understanding, fair, and confidential were qualities” and not “confidentiality”.

Line 115 should read better as “The relative youthfulness of the facilitators rather than “the relative youth of the social scientists”.

Line 117 In this statement “experience in administering FGDs and IDIs”, the word administering should be replaced by “facilitating” which is more appropriate.

Line 126-128In this statement “All caregivers were invited a week earlier and offered FGD attendance options based on their interest and availability.” The word interest is not quite clear. It may be best to remove “interest” and simply say “based on their availability”.

Reviewer #2: (No Response)

7. PLOS authors have the option to publish the peer review history of their article (what does this mean?). If published, this will include your full peer review and any attached files.

**Do you want your identity to be public for this peer review?** For information about this choice, including consent withdrawal, please see our Privacy Policy.

Reviewer #1: **Yes: **Tongdiyen Laura Jasper

Reviewer #2: **Yes: **Charles Martyn-Dickens

---

## [Editor Report · Decision Letter 3]

31 Aug 2023

Navigating antiretroviral adherence in boarding secondary schools in Nairobi, Kenya: a qualitative study of adolescents living with HIV, their caregivers and school nurses

PGPH-D-23-00355R3

Dear Mr Kipkurui,

We are pleased to inform you that your manuscript 'Navigating antiretroviral adherence in boarding secondary schools in Nairobi, Kenya: a qualitative study of adolescents living with HIV, their caregivers and school nurses' has been provisionally accepted for publication in PLOS Global Public Health.

Best regards,

Rashmi Josephine Rodrigues, M.D., Ph.D.

Academic Editor